# Triple-Negative Breast Cancer EVs Modulate Growth and Migration of Normal Epithelial Lung Cells

**DOI:** 10.3390/ijms25115864

**Published:** 2024-05-28

**Authors:** Ilaria Leone, Jessie Santoro, Andrea Soricelli, Antonio Febbraro, Antonio Santoriello, Barbara Carrese

**Affiliations:** 1IRCCS SYNLAB SDN, Via E. Gianturco, 80143 Naples, Italy; ilaria.leone@synlab.it (I.L.); andrea.soricelli@synlab.it (A.S.); barbara.carrese@synlab.it (B.C.); 2Oncology Unit, Casa di Cura Cobellis, Vallo della Lucania, 84078 Vallo della Lucania, Italy; antoniofebbraro@virgilio.it; 3Breast Unit, Casa di Cura Cobellis, Vallo della Lucania, 84078 Vallo della Lucania, Italy; antoniosantoriello2@gmail.com

**Keywords:** proliferation, triple-negative breast cancer, extracellular vesicles, intercellular communication, biomarkers

## Abstract

Breast cancer is the most common cancer amongst women worldwide. Recently, owing to screening programs and new technologies, the survival rate has increased significantly. Breast cancer can potentially develop metastases, and, despite them, lung metastases generally occur within five years of breast cancer diagnosis. In this study, the objective was to analyze the effect of breast cancer-derived EVs on a lung epithelial cell line. BEAS-2B cells were treated with extracellular vesicles (EVs) derived from triple-negative breast cancer cells (TNBCs), e.g., MDA-MB-231 and HS578T, separated using differential ultracentrifugation. We observed an increased growth, migration, and invasiveness of normal epithelial lung cells over time in the presence of TNBC EVs compared to the control. Therefore, these data suggest that EVs released by tumor cells contain biological molecules capable of influencing the pro-tumorigenic activity of normal cells. Exploring the role of EVs in oncology research and their potential cargo may be novel biomarkers for early cancer detection and further diagnosis.

## 1. Introduction

Breast cancer is one of the most common cancers in the world and mainly affects women, but occasionally it also occurs in men [1]. Many risk factors are associated with breast cancer, such as age, familiarity, and genetic mutations [2]. Other factors may contribute to breast cancer development, such as obesity or hormone therapies [3]. There are different breast cancer subtypes that mainly originate from epithelial cells. Therefore, it is possible to distinguish breast cancer as ductal carcinoma (70–80%); lobular carcinoma (10–15%); and other less frequent types, such as tubular, papillary, and mucinous carcinoma [4]. Breast cancer represents a heterogeneous group of tumors, and response to therapy is determined by biological features. The expression of estrogen receptor, progesterone receptor, and human epidermal growth factor receptor 2 allows us to determine the response to therapy [5]. Moreover, triple-negative breast cancers are characterized by the absence of hormone receptors and *HER2*, and they are categorized as the most aggressive subtype [6]. However, thanks to screening programs and new technologies, today the survival rate has increased significantly. Despite all of these advances in breast cancer diagnosis and treatment, there are no currently available treatments for this disease once it metastasizes to distant organs, including the lungs (21–32%), bones (30–60%), brain (4–10%), and liver (15–32%) [7]. Metastatic breast cancer cells that spread to organs with a soft microenvironment, such as the lung, have been shown to develop resistance to chemotherapy treatment in response to the enhancement of redox homeostasis [8]. In particular, lung metastases generally occur within five years of breast cancer diagnosis [9]. However, less than 1% of lung cancer instances are observed among men. These metastases alter normal lung physiology and are very difficult to treat. In fact, it is estimated that 60–70% of patients who die of breast cancer have lung metastasis [10]. In particular, the crosstalk between primary tumor and early formation of metastatic niches might be mediated by extracellular vesicles and tumor/stroma-derived factors [11]. EVs are membrane-enclosed vesicles released by all types of cells, and they carry various types of molecules, such as miRNA, proteins, lipids, and other bioactive factors. There is growing evidence that EVs recruit immune cells into the micro-niche of the lungs during chronic inflammatory diseases, including asthma and chronic obstructive pulmonary disorder. During this process, lung epithelial cells are exposed to inflammatory insults, and this damage induces structural re-arrangements of the lung epithelial cells, along with tissue remodeling, including increased metalloproteinase activity and altered cytokine levels [12]. Additionally, EVs are involved in the cellular communication between pulmonary epithelial cells and alveolar macrophages and form an important part of lung injury, including diffuse alveolar damage and lung epithelial cell death [13].

Nowadays, tumor-secreted extracellular EVs emerge among messengers in tumor progression and metastasis formation [14]. These EV cargoes alter recipient cell fates in local and distant tissues. Thus, EVs may be used as reliable biomarkers for early cancer detection and cancer development [15,16]. There is growing evidence that tumor-derived EVs can promote angiogenesis by regulating the activity of endothelial cells at distant secondary sites to facilitate metastasis [17]. In this regard, EVs’ cargo may be associated with cancer progression; in fact, recent studies showed that EVs induce epithelial mesenchymal transition (EMT) via the modulation of Hippo tumor suppressor signaling pathway in breast cancer [18]. Inactivation of the *Hippo* pathway triggers translocation of unphosphorylated *YAP/TAZ* to the nucleus and binds the *TEA* Domain Transcription Factor (*TEAD*) [19]. *TEAD*-mediated target genes modulate the expression of genes involved in EMT, such as Vimentin and N-cadherin (mesenchymal markers) and E-cadherin (epithelial markers) [20].

The objective of this study was to investigate how EVs derived from TNBC (HS578T and MDA-MB-231) can modulate cell growth and migration, while thriving from the invasive capacity of normal epithelial lung cells.

## 2. Results and Discussion

### 2.1. Characterization of EVs from TNBC and MCF10a Cell Lines by NTA and Immunoblotting Analysis

The particle concentration and size of EVs from breast cancer cell lines were characterized using NTA (Figure 1a and Appendix A). We observed that the particle concentration in HS578T EV samples was 5.4 × 10^16^ ± 3.5 × 10^15^ particles/mL and significantly (*p* < 0.0001) higher compared to MDA-MB-231 and MCF10a EVs samples, which had 2.8 × 10^15^ ± 2.5 × 10^14^ and 8.1 × 10^15^ ± 6 × 10^14^ particles/mL, respectively. Similarly, the particle size of HS578T cell-derived EVs was 150.6 ± 1.2 nm and significantly higher compared to MDA-MB-231 (*p* < 0.01) and MCF10a (*p* < 0.05) cells-EVs, which were 129.4 ± 3.3 nm and 134.6 ± 5.4 nm, respectively. Accordingly, data reported by others demonstrated that the number of EVs in breast cancer cell lines was higher compared to the non-malignant condition [21], suggesting that the EVs’ release by TNBC cells is much higher, as we reported in this study. 

Furthermore, the immunoblotting analysis supported the presence of EVs based on the detection of EVs as positive markers, following the MISEV2023 guidelines [15]. 

Specifically, three EVs-associated proteins were used, *TSG101*, *CD81*, and *CD63*; and one non-EVs-associated marker, calnexin, as reported in Figure 1b (and Appendix A). In the case of *TSG101*, we observed fainter bands in HS578T EVs and MCF10a EVs compared to MDA-MB-231 EVs and the control. Regarding *CD63*, it was homogeneously displayed in all the samples except for MCF10a EVs, for which we observed a weaker signal. Likewise, we detected clear bands for *CD81* but no signal in MCF10a EV samples. In addition, calnexin was not observed in any of the EV samples, but only in the cell lysate. Moreover, a group reported the expression of *TSG101* in MCF10a EVs, but no other markers were tested to confirm the presence of EVs [22].

Conversely, here, we used three positive EVs markers, as recommended by MISEV2023 guidelines [15]. Overall, our results demonstrated the presence of enriched EV samples, for which no contaminants were detected.

### 2.2. Cytoxicity Effect of EVs on Normal Human Bronchial Epithelial Cells

We examined the cytotoxic effect of EVs from TNBC and MCF10a on a BEAS-2B cell line (Figure 2). The effect was determined by MTS assay at 24, 48, and 72 h. BEAS-2B cells were treated with 10^8^ EVs derived from MDA-MB-231, HS578T, and MCF10a cell lines. Compared to the control (CTRL), no significant change was observed over time on cell viability when cells were incubated with EVs, regardless of the cell line. Therefore, no toxic effect was observed when BEAS-2B cells were treated with EVs isolated from MDA-MB-231, HS578T, and MCF10a cell lines, as reported in Figure 2. In this regard, it was reported in the literature that no effects were observed for the treatment with EVs from the MDA-MB-231 cell line on recipient cells after 48 h [23]. 

Using MTS assay, we measured cell viability by checking the cytotoxic effects of EVs derived from TNBC and MCF10a cell lines. We estimated the potential effect of EVs on BEAS-2B at different time points, which did not show any significant differences in all the conditions, as shown in Figure 2. Similarly, another study also showed the lack of toxic effects and changes in cell viability when treated with EVs [24]. These results confirm that the concentration of EVs used in this study did not alter cell viability or have a toxic effect on lung epithelial cells. 

### 2.3. Evaluation of EVs Uptake through Confocal Microscopy

To evaluate the uptake of PKH26-labeled cell-derived extracellular vesicles in recipient cells, a confocal microscopy analysis was performed. To this aim, BEAS-2B cells were incubated for 18 h with 10^8^ EVs separated from MDA-MB-231, HS578T, and MCF10a cell lines. In red are represented *β-actin* spindles, and in blue, the cell nuclei. As reported in Figure 3, confocal images show a co-localization of PHK26-labeled EVs (dotted yellow signals) towards the perinuclear area of normal human bronchial epithelial cells, where the cell structure was highlighted by the presence of *β-actin* (Figure 3). These results confirmed a cytoplasmic enrichment of PKH26-labeled EVs in a lung human bronchial epithelial cell line. In fact, EVs are usually taken up into endosomal compartments through endocytosis and require an operating cytoskeleton to induce an energy-dependent internalization process [25]. Therefore, we demonstrated that TNBC EVs are efficiently internalized in BEAS-2B cells (Figure 3). In addition, a similar concept was applied on MCF10a cells treated with MDA-MB-231 cell-derived EVs, which showed that EVs are internalized by non-malignant cells, as well as a possible modulation of their behavior [26]. Moreover, it was found that EVs are readily internalized by the *Raw264.7* cells [27]. There may be several pathways which regulate this mechanism of uptake, and further studies are required. Although, it was reported in the literature that lung cells take up MDA-MB-231 breast cancer cell-derived EVs mainly through macropinocytosis and clathrin-independent, caveolae-mediated endocytosis, and MEK inhibition [28]. However, here, using normal lung cells, we evaluated TNBC EVs’ uptake by recipient cells in order to mimic the potential effect of circulating EVs [29]. 

### 2.4. EVs Promotes Motility of Normal Human Bronchial Epithelial Cells

To investigate the effect of EVs isolated from MDA-MB-231, HS578T, and MCF10a cell lines on the motility of BEAS-2B cells, we performed a scratch-wound assay, as reported in Figure 4. We performed the wound-healing assay to measure the basic cell motility, and we observed how cells, at the edge of the created wound, migrate into the wound space (Figure 4a). 

To this aim, BEAS-2B cells were incubated for 24, 48, and 72 h with serum-free culture medium containing 10^8^ EVs separated from MDA-MB-231, HS578T, and MCF10a cell lines, after providing the wound. The stimulatory effect on cell motility was determined by evaluating the closure of the scratched area under starving conditions (Figure 4a). It is most reliably analyzed when performed using time-lapse imaging, which can also yield valuable cell morphology/protein localization information. Over time, cells treated with MDA-MB-231 EVs and HS578T EVs closed the wound much faster compared to all the other conditions, as shown in Figure 4a. BEAS-2B treated with TNBC EVs accelerated their migration rate considerably, already after 24 h (Figure 4a). Considering that cells were grown in the absence of FBS, we demonstrated that the stimulatory activity of BEAS-2B to migrate was mainly attributed to cancer cells’ EVs. Furthermore, as shown in Figure 4b, the rates of gap-filling in BEAS-2B cells treated with EVs derived from MDA-MB-231 and HS578T cell lines were significantly higher than the control group at 24 h (*p* < 0.0001), 48 h (*p* < 0.0001), and 72 h (*p* < 0.0001), which confirmed that cancer cells’ EVs promoted the migration of BEAS-2B cells (Figure 4b). Particularly, in absence of TNBC EVs, cells had slower scratch closure, and less cells migrated into the wound area. However, for the control and cells treated with MCF10a EVs, no significant differences were detected at 24 h (Figure 4b); meanwhile, at 48 h and 72 h, the wound was smaller (Figure 4a), with significant differences in % of closure rate at 48 h and 72 h (*p* < 0.05), as reported in Figure 4b. Likewise, after 48 and 72 h, the connection between BEAS-2B cells treated with TNBC EVs became closer compared to the control and MCF10a EVs’ treatment (Figure 4a). Similarly, another group established that, after 72 h, the cell density of HUVECs treated with EVs derived from MDA-MB-231 was significantly higher compared to the control [30]. Here, instead, we demonstrated the effect of TNBC EVs on the cell motility of normal lung cells, as such a result has not been published yet. Furthermore, a study on normal cells treated with MDA-MB-231 cell-derived EVs showed that the scratch area closed faster compared with the control [21]. In addition, it was demonstrated that co-culture experiments of HS578T cells with EVs from TNBC caused significant growth, a significant proliferation rate, and significant migration of the cells, confirming that EVs from TNBC cell lines can increase the invasiveness of the recipient cell [31]. 

Zhou et al. demonstrated that EVs purified from triple-negative MDA-MB-231 breast cancer cells promoted the metastasis-supporting behavior of primary human microvascular endothelial cells (HMVECs). They observed an increase in the migration and permeability of the endothelial layer in vitro that was attributed to the presence of miR-105 in EVs derived from MDA-MB-231 cells. An in vivo study showed that mice injected with MDA-MB-231 EVs showed similar pre-metastatic changes in lung endothelial cells, resulting in increased metastasis to the lung [32]. 

Therefore, in line with the literature, our study demonstrated that TNBC EVs significantly increased the cell motility of normal human bronchial epithelial cells when treated with triple-negative breast cancer EVs, s result that was not observed in the case of EVs derived from the MFC10a cell line.

Altogether, our results suggest potential molecular signatures of the studied TNBC cell lines and their impact on growth and proliferation rates [33]. 

### 2.5. BEAS-2B Cells Have Higher Migration Rate in Presence of Breast Cancer-Derived EVs 

Additionally, to evaluate the migration of normal human bronchial epithelial cells, BEAS-2B cells treated with EVs separated from MDA-MB-231, HS578T, and MCF10a cell lines up to 72 h and were tested using Transwell migration assay (Figure 5). Compared to control cells, an improved cell migration through the transwell chamber was observed when cells were incubated with EVs separated from both triple-negative cell lines (MDA-MB-231 and HS578T), in line with the results observed with the scratch assay. In fact, in Figure 5, we observed that the migrated cells’ ratio was significantly higher after 24 h (*p* < 0.0001), 48 h (*p* < 0.01), and 72 h (*p* < 0.0001) when BEAS-2B cells were treated with breast cancer-derived EVs compared to the control. Likewise, the same results were confirmed by the increased crystal violet eluted from migrated cells. Furthermore, as reported in Appendix A, representative confocal images show increased BEAS-2B cell migration through the transwell chamber up to 72 h when incubated with EVs from TNBC cell lines, compared to the control and cells treated with EVs separated from MCF10a cells. Moreover, a reduction in the number of non-migrated cells on the upper side of the transwell chamber was also observed at 72 h, which confirmed the increased migration of BEAS-2B cells after treatment with TNBC EVs. In addition, these results showed the same trend observed in the wound-healing assay, where TNBC EVs significantly increased the motility of BEAS-2B. In this regard, it was reported that EVs derived from TNBC cells had a higher expression of two miRNAs (miR-185-5p and miR-652-5p), known to be involved in cell migration, compared to EVs from MCF10a cells, which suggest a more aggressive effect of TNBC EVs [34]. Additionally, MDA-MB-231-derived EVs increased the migration number and migration ability of HUVECs compared with controls [30]. Furthermore, it has been shown that cancer EVs are capable of leading cancer progression by transferring biological traits from their tumor of origin [16]. Despite the underlying mechanism of TNBC, EVs still need further investigation; this study supports the fact that TNBC EVs might transport biological materials which facilitate TNBC progression.

The effect of breast cancer EVs does not exclude their carcinogenic potential, confirming that our non-toxic EVs derived from MDA-MB-231, HS578T, and MCF10a cell lines, as reported in the MTS assay, can trigger the motility of normal human bronchial epithelial cells. 

## 3. Materials and Methods

### 3.1. Cell Lines

Human bronchial epithelial cell line (BEAS-2B), triple-negative breast tumor cell lines (HS578T and MDA-MB-231), and mammary breast fibrocystic disease cell line (MCF10a) cells were grown in Dulbecco’s modified Eagle medium (DMEM, GIBCO, Waltham, MA USA) supplemented with 10% heat-inactivated FBS (GIBCO, Waltham, MA USA), 100 U/mL penicillin, 100 mg/mL streptomycin, and 1% L-glutamine. Cells were grown at 37 °C in a 5% CO_2_ atmosphere. All cells used in this study were obtained from IRCCS Synlab SDN Biobank [35] (https://doi.org/10.5334/ojb.26, accessed on 19 May 2024).

### 3.2. EVs Separation from Breast Cancer Cells

EVs from MDA-MB-231, HS578T, and MCF10a cell lines were separated by differential ultracentrifugation after 48 h in culture medium exo-free. Next, medium from all cell lines was centrifuged at 300× *g* for 10 min to remove cells. Then, the supernatant was centrifuged at 2000× *g* for 10 min to remove dead cells. Larger EVs and remaining cells debris were removed by ultracentrifugation at 10,000× *g* for 30 min at 4 °C, using OPTIMA MAX-XP (Cat: # 393315, Beckman Coulter, Brea, CA, USA). Cleared conditioned medium was then ultracentrifuged at 100,000× *g* for 70 min at 4 °C for pelleting EVs. Then, EV pellets were washed using 0.22 µm filtered phosphate saline buffer (PBS) at 100,000× *g* for 70 min. Finally, enriched EV samples were resuspended in 100 µL of 0.22 µm filtered PBS and used immediately for further analysis. Prior to the EVs’ separation, supplemented FBS was centrifuged for 18 h at 118,000× *g* at 4 °C for EVs’ depletion, following previous publication [36]. Triple-negative breast cancer and MCF10a cell-derived EVs were isolated by ultracentrifugation as previously described [37].

### 3.3. Particle Concentration and Size Using Nanoparticle Tracking Analysis (NTA)

The concentration and sizes of particles in the EVs derived from MDA-MB-231, HS578T, and MCF10a cell lines were analyzed using NTA (NanoSight NS300, Malvern Instruments Ltd., Malvern, UK). EV samples were automatically injected into the NTA system under constant flow conditions (flow rate = 50). A minimum of five × 60 s videos were recorded and were analyzed using NTA 3.2 software (Malvern Instruments Ltd., Malvern, UK). The detection threshold during analysis was selected to ensure that only distinct nano-objects were analyzed and that any artefacts were removed. Three replicates of each sample were analyzed by NanoSight NS300, independently. NTA analysis was performed following previous publication [38]

### 3.4. Immunoblotting Analysis of EVs

Further characterization of EVs was performed using immunoblotting analysis. Essentially, EVs were lysed using *JS lysis buffer* (HEPES 1 M; NaCl 5 M; Glicerol 100%; Triton X100; MgCl2 1 M; EGTA 0.1 M; H2O). After that, lysed EVs (30 μg) were resolved on 10% Mini-PROTEAN^®^ TGX Stain-Free™ (Bio-Rad Laboratories, Hercules, CA, USA, Cat. #4568034) at 120V, and proteins were transferred by the Trans-Blot Turbo System (Bio-Rad Laboratories, Cat. # 690BR024275). Filters were blocked with 5% milk in TBST containing 0.1% Tween-20 for 1 h and incubated overnight at 4 °C with primary antibodies. Specifically, we used anti-TSG101 (1:1000; Cat. # ab30871), anti-CD81 (1:500; Cat. # sc-116029, Santa Cruz Biotechnology, Inc., Dallas, TX, USA), anti-CD63 (1:1000; Cat. # ab68418), and anti-calnexin (1:1000; Cat. # ab10286, Abcam, Cambridge, UK). The secondary antibodies used were Goat Anti-Mouse IgG (1:2000; Cat. # 10303-05, Biotech, Alpharetta, GA, USA) or Goat Anti-Rabbit IgG (1:2000; Cat. # 4030-05, Biotech). Imaging was performed using an automated ChemiDoc™ MP Imaging System (Bio-Rad Laboratories, Cat. # 12003154, Segrate, MI, Italy) and Clarity Max™ Western ECL Substrate (Cat. # 1705062, Bio-Rad Laboratories Segrate, MI, Italy). BEAS-2B cell lysate (CL) was used as a positive control. Immunoblotting analysis was performed based on a study previously published by our group [39].

### 3.5. Cytotoxicity Assay (MTS Assay)

To test the toxicity effect of EVs, MTS [3-(4,5-dimethylthiazol-2-yl)-5-(3-carboxymethoxyphenyl)-2-(4-sulfophenyl)-2H tetrazolium] assay was performed (Cat. # G3580, CellTiter 96 Aqueous One Solution Cell Proliferation Assay, Promega, Milano, Italy). For the cytotoxicity assay, BEAS-2B cells were seeded at a density of 1.5 × 10^3^ per well in 96-well plates and incubated with 10^8^ EVs separated from MDA-MB-231, HS578T, and MCF10a cell lines for 24, 48, and 72 h, following the manufacturer’s instructions. At each time point, absorbance was recorded at 490 nm by a Multilabel Reader (Victor Nivo Multimode Microplate Reader, PerkinElmer, Waltham, MA, USA).

### 3.6. Uptake of EVs Analyzed by Immunofluorescence

EVs separated from MDA-MB-231, HS578T, and MCF10a cell lines were labeled with the yellow fluorescent cell membrane linker PKH26 (Cat. #MIDI26-1KT, Sigma-Aldrich, St. Louis, MO, USA). Briefly, 10^8^ EVs from each breast cancer cell line were used and stained with PHK26 dye solution for 15 min at 37 °C. Next, labeled EVs were pelleted at 100,000× *g* for 70 min at 4 °C, using the ultracentrifuge OPTIMA MAX-XP (Cat: # 393315, Beckman Coulter, USA). To evaluate the uptake of fluorescent PKH26-EVs, a confocal microscopy analysis was performed. To this aim, BEAS-2B cells were seeded in µ-Slide 8 Well high (Cat. # 80841 IBIDI GMBH, Gräfelfing, Germany) at a density of 4 × 10^4^ and incubated for 18 h at 37 °C, with 10^8^ EVs/mL separated from MDA-MB-231, HS578T, and MCF10a cell lines. Cells were fixed and permeabilized with cold methanol (100%) for 20 min at −20 °C. Cell nuclei were stained with DAPI (Cat. # D1306, Invitrogen) at 1:35,000 for 15 min at room temperature (RT), in dark conditions. After three washes in PBS 1×, cells were blocked in 1% Bovine Serum Albumin (BSA) (Cat. # A1391, PanReac AppliChem, Monza, Italy) solution at RT for 1 h. Subsequently, cells were stained with anti-β-actin primary antibody (1:50, Cat. # AF0342, Bioscience) for 1 h at RT. After two washes in PBS 1×, the secondary antibody, Goat Anti-Mouse IgG (H+L) Cross-Adsorbed Secondary Antibody Alexa Fluor™ 750 (1:200, Cat. # A21037, Thermo Fisher, Waltham, MA, USA) was added for 30 min at RT. Images were acquired by confocal microscopy (Mica, Leica Microsystems, Wetzlar, Germany) with a 63× magnification and saved in TIFF format. Samples used for this experiment were prepared based on a previous publication, with some modifications [40].

### 3.7. Scratch Assays

BEAS-2B cells were plated at a density of 3 × 10^4^ in a 12-well plate (Cat. # 3513, Corning, Corning, NY, USA), and, when confluence was reached, the culture medium was removed, and a scratched wound was performed with 200 µL tips in each well. Then, cells were washed in PBS. Next, cells were grown in DMEM culture medium serum-free; implemented with 1% penicillin/streptomycin and 1% L-glutamine; and treated with 10^8^ EVs derived from MDA-MB-231, HS578T, and MCF10a cell lines and PBS solution for the control. Images were acquired by confocal microscopy with a 10× magnification in different fields of the wound at the scratch moment (0 h) and after 24 h, 48 h, and 72 h. Images were saved in TIFF format. The scratch area was calculated with ImageJ software (https://imagej.net/ij/, accessed on 19 May 2024) and analyzed to measure the wound-healing ability of BEAS-2B cells treated with EVs, following a previous publication [30]. 

### 3.8. Transwell Migration Assay

The migration assay was performed with Permeable Support for 24-well Plate with 1.0 µm Transparent PET Membrane (Cat. # 353104, Falcon). BEAS-2B cells were seeded at a density of 4 × 10^4^ cells in a 24-well plate and pretreated with 10^8^ EVs, separated from MDA-MB-231, HS578T, and MCF10a cell lines, for 24 h. For migration assay, 500 cells were seeded in DMEM FBS-depleted in the upper part of the Transwell chamber, as previously described [30]. The lower part of the chamber was filled with 600 mL of DMEM medium supplemented with 10% FBS to induce the cells’ migration. BEAS-2B cells were incubated at 37 °C for 24, 48, and 72 h. Then, the chambers were stained with 0.1% crystal violet in 25% methanol for 20 min. The chambers were accurately washed in water, and non-migrated cells were removed with a cotton swab. Crystal violet was eluted with 600 mL of 1% SDS-PBS for each well, and absorbance was recorded at 590 nm by a Victor Nivo Multimode Microplate Reader. Representative images were obtained fixing cells with cold methanol (100%) for 20 min at −20 °C. Cell nuclei were stained with DAPI at 1:35,000 for 15 min at RT, in the dark. Images were acquired by confocal microscopy with a 10× magnification and saved in TIFF format.

### 3.9. Statistical Analysis

Data are expressed as mean ± SEM of three independent experiments. The statistical significance of differences among groups was evaluated using One-Way ANOVA test analysis through the software GraphPad Prism 9.5.1. We submitted all relevant data from our experiments to the EV-TRACK knowledgebase (EV-TRACK ID: EV231023) [41].

## 4. Conclusions

TNBC is a particularly aggressive subtype of breast cancer, with an earlier onset of metastatic disease and rapid progression. Currently, there is an urgent need to develop early diagnosis tools, such as EVs. Our preliminary data highlight some mechanisms adopted by EVs to characterize the oncogenic phenotype of breast cancer and the effect of TNBC EVs on normal epithelial lung cells. Based on that, we demonstrated an increased rate of cell growth, migration, and invasion after EVs’ internalization in comparison with the healthy control (EVs from MCF10a cells) and in the absence of EVs. This study is the first of its kind where lung epithelial cells are treated with TNBC-derived EVs. Our data provide the basis for studying the mechanism of TNBC EVs and their function in promoting cell motility and invasiveness of recipient cells. Although there is still a lot to discover on tumor progression and its related features, our data suggest a potential system of TNBC EVs’ activity and pave the way for future studies offering an in-depth analysis of TNBC EVs. 

## Figures and Tables

**Figure 1 ijms-25-05864-f001:**
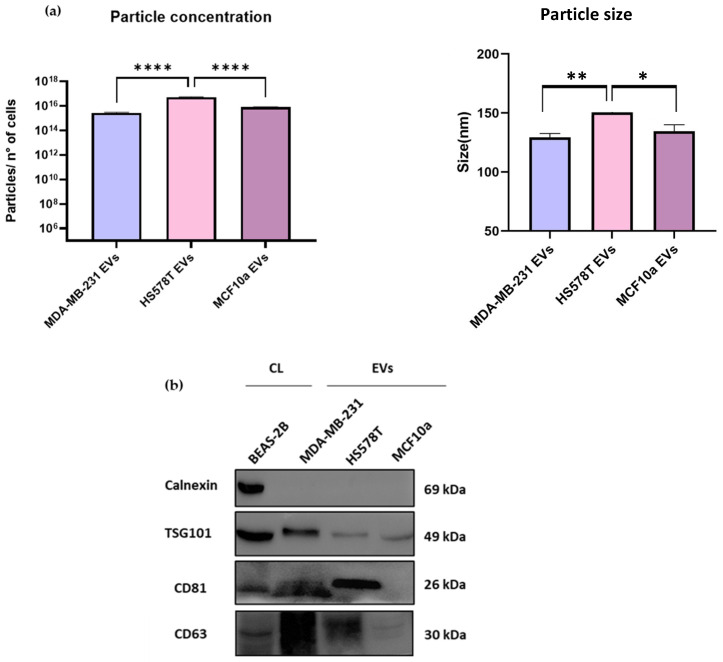
Analysis of particle concentration and size using NTA and immunoblotting analysis of EV markers of TNBC EVs and MCF10a-derived EVs. (**a**) Nanoparticle tracking analysis of EVs. Particle sizes and concentration analysis from MDA-MB-231, HS578T and MCF10a cell-derived EVs. A minimum of five × 60 s videos was recorded for each sample. Three replicates of each sample were analyzed by NTA independently. Data are presented as mean bars ± SEM. The p-values were calculated using One-Way ANOVA test (* *p* < 0.05, ** *p* < 0.01, **** *p* < 0.0001). (**b**) Immunoblotting analysis of separated EVs from TNBC and MCF10a cells. Analysis of EVs-associated markers TSG101, CD81, and CD63 and negative marker calnexin.

**Figure 2 ijms-25-05864-f002:**
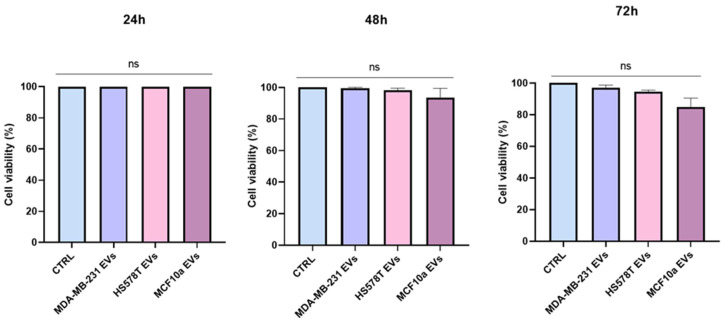
Cytoxicity effect of EVs on normal lung epithelial cells analyzed by MTS assay. MTS assay of BEAS-2B cells treated with MDA-MB-231, HS578T, MCF10a cell-derived vesicles, or PBS (CTRL) for 24, 48, and 72 h. Standard deviations were calculated on replicates from three independent experiments. The *p*-values were calculated using One-Way ANOVA test (ns = not significant).

**Figure 3 ijms-25-05864-f003:**
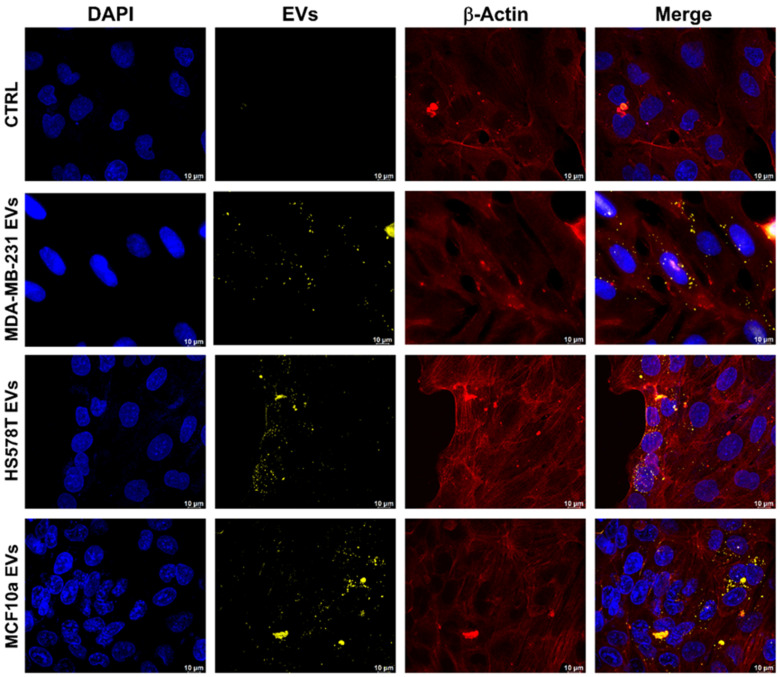
Confocal microscopy of cellular uptake of PKH26-labeled TNBC EVs and MCF10a EVs by BEAS-2B cell line. Maximum projection images of BEAS-2B cells incubated with 10^8^ EVs separated from MDA-MB-231, HS578T, and MCF10a cell lines for 18 h. Scale bar: 10 µm. CTRL = PBS treated.

**Figure 4 ijms-25-05864-f004:**
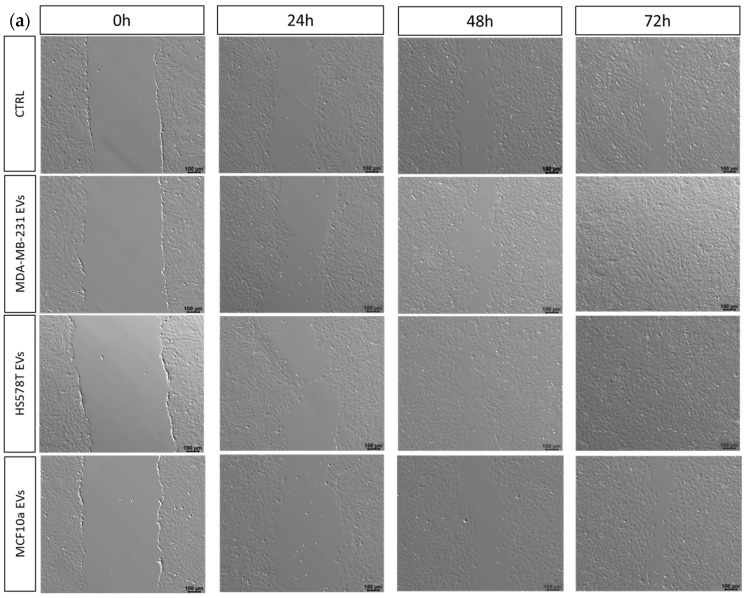
Wound-healing assay of BEAS-2B cells treated with TNBC EVs and MCF10a EVs. (**a**) Representative wound-healing images of BEAS-2B cells treated with 10^8^ MDA-MB-231, HS578T, and MCF10a cell line-derived EVs after 0, 24, 48, and 72 h. Scale bar: 100 µm. (**b**) Relative wound closure rate of BEAS-2B cells treated with 10^8^ MDA-MB-231, HS578T, and MCF10a cell line-derived EVs after 24, 48, and 72 h. Standard deviations were calculated on replicates from three independent experiments. The *p*-values were calculated using One-Way ANOVA test (* *p* < 0.05, **** *p* < 0.0001, ns = not significant).

**Figure 5 ijms-25-05864-f005:**
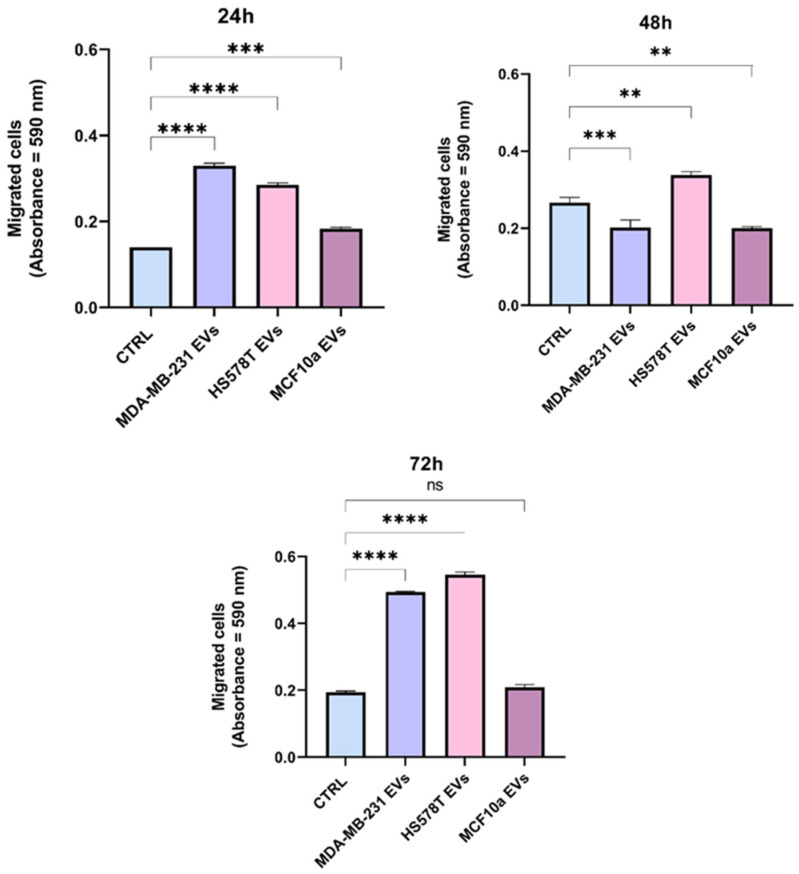
Analysis of BEAS-2B cell migration in presence of TNBC and MCF10a cell-derived EVs using Transwell migration assay. Migration assay of BEAS-2B cells cultured in the presence of MDA-MB-231, HS578T, MCF10a cell-derived vesicles or PBS (CTRL) for 24, 48 and 72 h. SEM were calculated on replicates from three independent experiments. The p values were calculated using One-way Anova test (** *p* < 0.01, *** *p* < 0.001, **** *p* < 0.0001, ns = not significant).

## Data Availability

Raw data generated in this study can be available. Please contact the corresponding author, J.S. (jessie.santoro@synlab.it).

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
