# Peer review of "Triple-Negative Breast Cancer EVs Modulate Growth and Migration of Normal Epithelial Lung Cells"

_ijms, 2024, doi:10.3390/ijms25115864_

Round 1

Reviewer 1 Report (Previous Reviewer 1)

Comments and Suggestions for Authors

Breast cancer is one of the most common cancer types observed in women's worldwide population (with only approximately less than 1% of men affected). Despite increased diagnostic capabilities and therapeutic schemes, breast cancer and its metastasis to the lungs, bones, brain, and liver remains a huge problem for current oncology. The article titled “Triple Negative Breast Cancer EVs Modulate Growth and 2 Migration of Normal Epithelial Lung Cells” by Leone I. et al. focuses on TNBC-derived EVs impact on normal lung cell physiology and its potential connotation with preparation of the pre-metastatic niche for breast cancer metastasis. The authors addressed this issue by isolating EVs from two TNBC (MDA-MB-231 and HS578T) cell lines and analyzing their impact on BEAS-2B cell line migration. This is an interesting subject, and the current version of the manuscript is corrected according to my previous concerns.

Author Response

Reviewer 1

Top of Form

Open Review

Quality of English Language

( ) I am not qualified to assess the quality of English in this paper 
( ) English very difficult to understand/incomprehensible 
( ) Extensive editing of English language required 
( ) Moderate editing of English language required 
( ) Minor editing of English language required 
(x) English language fine. No issues detected 

Yes

Can be improved

Must be improved

Not applicable

Does the introduction provide sufficient background and include all relevant references?

(x)

( )

( )

( )

Are all the cited references relevant to the research?

( )

(x)

( )

( )

Is the research design appropriate?

(x)

( )

( )

( )

Are the methods adequately described?

(x)

( )

( )

( )

Are the results clearly presented?

(x)

( )

( )

( )

Are the conclusions supported by the results?

(x)

( )

( )

( )

Comments and Suggestions for Authors

Breast cancer is one of the most common cancer types observed in women's worldwide population (with only approximately less than 1% of men affected). Despite increased diagnostic capabilities and therapeutic schemes, breast cancer and its metastasis to the lungs, bones, brain, and liver remains a huge problem for current oncology. The article titled “Triple Negative Breast Cancer EVs Modulate Growth and 2 Migration of Normal Epithelial Lung Cells” by Leone I. et al. focuses on TNBC-derived EVs impact on normal lung cell physiology and its potential connotation with preparation of the pre-metastatic niche for breast cancer metastasis. The authors addressed this issue by isolating EVs from two TNBC (MDA-MB-231 and HS578T) cell lines and analyzing their impact on BEAS-2B cell line migration. This is an interesting subject, and the current version of the manuscript is corrected according to my previous concerns.

We thank the reviewer for his/her thoughtful comments.

Submission Date

24 April 2024

Date of this review

29 Apr 2024 14:04:04

Bottom of Form

Reviewer 2 Report (Previous Reviewer 2)

Comments and Suggestions for Authors

After revision, the manuscript still has considerable shortcomings that require further attention and improvement.

1. Limited exploration of mechanisms: A notable limitation in the manuscript is the lack of exploration of the mechanisms by which extracellular vesicles influence physiological changes in lung epithelial cells. Understanding these mechanisms is critical to underpin the scientific value of the data presented, as it would provide insight into the potential pathways and interactions at play. I recommend the inclusion of studies or experiments that could shed light on these underlying mechanisms, possibly through molecular biology techniques.

2. Lack of a discussion section: the manuscript currently lacks a section discussing the implications of the results for the existing literature in the field. It is important that you situate your results within the broader scientific dialog, particularly in relation to "pro-tumorigenic activity of normal cells". A well-rounded discussion would not only increase the scientific depth of the article, but also provide a clearer understanding of where your results stand in comparison to other studies, potentially highlighting both similarities and discrepancies. The authors' conclusion, "Therefore, in line with others, our study report that TNBC EVs significantly incremented growth and proliferation of normal human bronchial epithelial cells when treated with breast cancer EVs which was not observed in the case of EVs derived from MFC10a cell line," contradicts the results from Section 2.3, "Cytotoxic effect of EVs on Normal Human Bronchial Epithelial Cells." In this section, the MTT assay showed no effect of EVs on the proliferation of the normal bronchial epithelial cell line. These contradictions require clarification.

In several parts of the manuscript, including the abstract, you emphasize the potential of using extracellular vesicles as markers for disease. However, I noticed that there is no experimental data or detailed exploration of this aspect in the manuscript. The lack of a focused study or discussion on the use of extracellular vesicles as a disease marker seems to be a disconnect between what is being proposed and the research that has been done.

Author Response

Reviewer 2

Open Review

Quality of English Language

(x) I am not qualified to assess the quality of English in this paper 
( ) English very difficult to understand/incomprehensible 
( ) Extensive editing of English language required 
( ) Moderate editing of English language required 
( ) Minor editing of English language required 
( ) English language fine. No issues detected 

Yes

Can be improved

Must be improved

Not applicable

Does the introduction provide sufficient background and include all relevant references?

( )

(x)

( )

( )

Are all the cited references relevant to the research?

(x)

( )

( )

( )

Is the research design appropriate?

( )

( )

(x)

( )

Are the methods adequately described?

(x)

( )

( )

( )

Are the results clearly presented?

( )

(x)

( )

( )

Are the conclusions supported by the results?

( )

( )

(x)

( )

Comments and Suggestions for Authors

After revision, the manuscript still has considerable shortcomings that require further attention and improvement.

  1. Limited exploration of mechanisms: A notable limitation in the manuscript is the lack of exploration of the mechanisms by which extracellular vesicles influence physiological changes in lung epithelial cells. Understanding these mechanisms is critical to underpin the scientific value of the data presented, as it would provide insight into the potential pathways and interactions at play. I recommend the inclusion of studies or experiments that could shed light on these underlying mechanisms, possibly through molecular biology techniques.

We thank the reviewer for his/her comments.  In literature, other studies have explored the effect of EVs on morphological changes in lung epithelial cells. In fact, the interaction between EVs and pulmonary epithelial cells induce morphological changes in cells and promotes migration processes as we have also demonstrated with our data. REF:Extracellular vesicles from mast cells induce mesenchymal transition in airway epithelial cells (Yin et al. Respiratory Research (2020) 21:101 https://doi.org/10.1186/s12931-020-01346-8)

In addition, EVs from several cellular sources, in the microenvironment influence the pathological process in the lung through different pathways. In particular, in this review REF:Multiple Biological Roles of Extracellular Vesicles in Lung Injury and Inflammation Microenvironment (Biomed Res Int. 2020; 2020: 5608382. Published online 2020 Jul 14. doi: 10.1155/2020/56083829, they investigated the mechanism of crosstalk between lung epithelial cells and immune cells through extracellular vesicles during lung inflammation.
Althoug, there is a body of knowledge regarding the interaction of cancer EVs in the cellular environment, we aim to investigate multiple pathways directly controlled by interaction with EVs in the context of breast cancer and possible lung metastasis formation. The introduction of the revised manuscript include studies and experiments found in literature which underline the mechanisms of morphological influence of EVs in lung epithelial cells. All the changes are reported in red.

  1. Lack of a discussion section: the manuscript currently lacks a section discussing the implications of the results for the existing literature in the field.

We thank the reviewer for his/her comments. We decided to structure the paper by combining the results and discussion part to make it easier to read and understand the data. Our approach was to simultaneously have the commentary and interpretation of the results, while avoiding any difficulties for the readers. All the changes are reported in red.

It is important that you situate your results within the broader scientific dialog, particularly in relation to "pro-tumorigenic activity of normal cells". A well-rounded discussion would not only increase the scientific depth of the article, but also provide a clearer understanding of where your results stand in comparison to other studies, potentially highlighting both similarities and discrepancies.

We thank the reviewer for his/her comment. We improved our results section by including a wider discussion of data. However, how and by which how and by which mechanisms EVs can regulate the behavior of tumor cells and non-cancer cells is at the center of debate. Newer research has shown that EVs are capable of transforming normal cells into cancer cells. In fact, there is a growing evidence that proteins contained within breast cancer derived EVs can modify the behaviour of endothelial cells and develop various cancer processes such as migration, proliferation, angiogenesis, and extracellular matrix degradation REF: https://doi.org/10.1158/1541-7786. Therefore, our study and work already done by other would greatly improve our understanding of the molecular pathways involved in cancer biology.

The authors' conclusion, "Therefore, in line with others, our study report that TNBC EVs significantly incremented growth and proliferation of normal human bronchial epithelial cells when treated with breast cancer EVs which was not observed in the case of EVs derived from MFC10a cell line," contradicts the results from Section 2.3, "Cytotoxic effect of EVs on Normal Human Bronchial Epithelial Cells." In this section, the MTT assay showed no effect of EVs on the proliferation of the normal bronchial epithelial cell line. These contradictions require clarification.

We thank the reviewer for his/her comment. We have changed the line "Therefore, in line with others, our study report that TNBC EVs significantly incremented growth and proliferation of normal human bronchial epithelial cells when treated with breast cancer EVs which was not observed in the case of EVs derived from MFC10a cell line" with “Therefore, in line with the literature, our study demonstrated that TNBC EVs signifi-cantly increased cell motility of normal human bronchial epithelial cells when treated with triple negative breast cancer EVs which was not observed in the case of EVs derived from MFC10a cell line”, in the revised version of the manuscript.

contradicts the results from Section 2.3, "Cytotoxic effect of EVs on Normal Human Bronchial Epithelial Cells." In this section, the MTT assay showed no effect of EVs on the proliferation of the normal bronchial epithelial cell line. These contradictions require clarification.

The contradictions that emerged were clarified by emphasizing the effect of EVs on cell motility. The purpose of the MTS assay was only to investigate the cytotoxic effect of TNBC and MCF10a EVs on lung epithelial cells. Specifically, the MTS assay is used to measure cell metabolic activity as an indicator of cell viability but also cytotoxicity. This colorimetric assay is based on the reduction of MTT to purple formazan crystals by metabolically active cells. On the other side, the wound healing assay is a standard in vitro technique to probe cell motility. In this assay, we used culture medium serum-free to inhibit cell proliferation and to observe the wound closure occuring mainly in the cells treated with TNBC EVs.

For this reason, we conducted both MTT and wound healing assay under starvation conditions.
Therefore, the MTT assay measures cell viability and is mainly used to test the cytotoxic effects of the compounds under study. The wound healing assay measures basic parameters of cell migration, such as speed, persistence and polarity.

In several parts of the manuscript, including the abstract, you emphasize the potential of using extracellular vesicles as markers for disease. However, I noticed that there is no experimental data or detailed exploration of this aspect in the manuscript. The lack of a focused study or discussion on the use of extracellular vesicles as a disease marker seems to be a disconnect between what is being proposed and the research that has been done.

We thank the reviewer for his/her comment. We are currently working on empthazing this aspect of EVs using multi-omics approach, such as proteonimcs. We hope to unravel new biomarkers which could explain the differences between triple negative breast cancer EVs and non malignant cell derived-EVs on cell motility. The full characherization of breast cancer EVs could substantially advance the early detection of tumors and introduce novel approaches for tumor management. The diagnosis of tumors at an early stage and/or to monitor the progression of established tumors.

Growing body of evidence indicates that EVs may also serve as innovative diagnostic biomarkers for various types of cancer REF: https://link.springer.com/article/10.1186/s12943-024-01932-0#Sec3. It has been shown that EVs collected from breast cancer patients have different molecular contents compared to healthy donor REF: https://www.ncbi.nlm.nih.gov/pmc/articles/PMC10380344/.

Reviewer 3 Report (New Reviewer)

Comments and Suggestions for Authors

Summary:

In this original paper, Leone Ilaria et al., investigated how extracellular vesicles (EVs) derived from triple negative breast cancer cells (TNBC) (HS578T and MDA-MB-231) can modulate cell growth and migration, while thriving invasive capacity of normal epithelial lung cells. Overall, the paper is very well written and the material and method section is well described. The authors provided the raw data, which I think is very important nowadays, as it gives more credibility to the results presented.The result section is well presented, and the experiments were well designed to prove the authors' hypothesis. 

Comments:

The manuscript cover an interesting topic in oncobiology. The introduction is well organized and easy to read and follow. However, regarding materials and methods, I suggest the authors describe in more detail the methodology carried out for each experiment. In addition, if the methods used have been previously published by the same group or by other research groups, please, at the end of each methodology, add the reference of the original paper.

In the wound assay, did the authors use any cell proliferation inhibitors? Do not the authors think that cell proliferation can mask cell motility in this type of assay?

Figure 2 is missing. I couldn't find it in the manuscript

I suggest the authors rewrite the captions for figures 1, 3, 4, 5 and 6. The figures present a lot of information, but the captions are poor in content. They need to be rewritten in more detail.

Author Response

Reviewer 3

Top of Form

Bottom of Form

Top of Form

Review Report Form 

Open Review

Quality of English Language

( ) I am not qualified to assess the quality of English in this paper 
( ) English very difficult to understand/incomprehensible 
( ) Extensive editing of English language required 
( ) Moderate editing of English language required 
( ) Minor editing of English language required 
(x) English language fine. No issues detected 

Yes

Can be improved

Must be improved

Not applicable

Does the introduction provide sufficient background and include all relevant references?

(x)

( )

( )

( )

Are all the cited references relevant to the research?

(x)

( )

( )

( )

Is the research design appropriate?

(x)

( )

( )

( )

Are the methods adequately described?

( )

(x)

( )

( )

Are the results clearly presented?

( )

(x)

( )

( )

Are the conclusions supported by the results?

(x)

( )

( )

( )

Comments and Suggestions for Authors

Summary:

In this original paper, Leone Ilaria et al., investigated how extracellular vesicles (EVs) derived from triple negative breast cancer cells (TNBC) (HS578T and MDA-MB-231) can modulate cell growth and migration, while thriving invasive capacity of normal epithelial lung cells. Overall, the paper is very well written and the material and method section is well described. The authors provided the raw data, which I think is very important nowadays, as it gives more credibility to the results presented.The result section is well presented, and the experiments were well designed to prove the authors' hypothesis. 

Comments:

The manuscript cover an interesting topic in oncobiology. The introduction is well organized and easy to read and follow. However, regarding materials and methods, I suggest the authors describe in more detail the methodology carried out for each experiment. In addition, if the methods used have been previously published by the same group or by other research groups, please, at the end of each methodology, add the reference of the original paper.

We thank the reviewer for his/her thoughtful comments. We improved the material and method section in this study and we added all the references at the end of each methodology, as suggested.

In the wound assay, did the authors use any cell proliferation inhibitors? Do not the authors think that cell proliferation can mask cell motility in this type of assay?

We thank the reviewer for his/her comment. We performed the scratch assay using starving conditions. In fact cells were incubated with culture medium serum-free and then treated with EVs. We demonstrated how the cell motility of BEAS-2B is driven primarily by TNBC EVs comparing to the control and MCF10a EVs. All the changes are reported in red.

Figure 2 is missing. I couldn't find it in the manuscript

We apologise for this mistake. We numbered the figures correctly from figure 1 to figure 5. All the changes are reported in red.

I suggest the authors rewrite the captions for figures 1, 3, 4, 5 and 6. The figures present a lot of information, but the captions are poor in content. They need to be rewritten in more detail.

We thank the reviewer for his/her comments. We changed all the figure captions by adding more details, as suggested. All the changes are highlighted in red.

Submission Date

24 April 2024

Round 2

Reviewer 2 Report (Previous Reviewer 2)

Comments and Suggestions for Authors

The authors have successfully dealt with all the comments and made the necessary revisions to the manuscript. The previously identified contradictions in the discussion of the data have been resolved, significantly improving the arguments and increasing the scientific credibility of the work. The revised version of the article shows an improved quality of presentation, logical coherence and a high level of scientific analysis. In view of the extensive revisions, the article is now ready for publication. I recommend acceptance of the article without further revisions.

Reviewer 3 Report (New Reviewer)

Comments and Suggestions for Authors

I thank the authors for the changes made

This manuscript is a resubmission of an earlier submission. The following is a list of the peer review reports and author responses from that submission.

Round 1

Reviewer 1 Report

Comments and Suggestions for Authors

Breast cancer is one of the most common cancer types observed in women's worldwide population (with only approximately less than 1% of men affected). Despite increased diagnostic capabilities and therapeutic schemes, breast cancer and its metastasis to the lungs, bones, brain, and liver remains a huge problem for current oncology. The article titled “Triple Negative Breast Cancer EVs Modulate Growth and 2 Migration of Normal Epithelial Lung Cells” by Leone I. et al. focuses on TNBC-derived EVs impact on normal lung cell physiology and its potential connotation with preparation of the pre-metastatic niche for breast cancer metastasis. The authors addressed this issue by isolating EVs from two TNBC (MDA-MB-231 and HS578T) cell lines and analyzing their impact on BEAS-2B cell line migration. This is an interesting subject, however, in my opinion, the current version requires substantial revision, as it seems to present rather preliminary data and lacks some important parts (and possibly some concluding experiments) to be published by IJMS.

My concerns can be distinguished into major and minor.

Major issues:

1)     In my opinion this manuscript lacks proper discussion of the obtained results. The “Results and Discussion” section mainly focuses on the acquired results but not on their meaning. No mechanism for increased migration of normal lung cells is proposed, absence of CD81 and CD63 in the EVs isolated from MCF10a cells is not discussed at all. Statements such as: (LINE 289-292) “We believe that our data will help in understanding the transition to the metastatic stage. We provide evidence for the use of extracellular vesicles as early biomarker detectable in several fluids with the objective of predicting metastasis” which are not backed by the acquired data nor discussed. Increased migration of normal lung cells does not directly contribute to the increased probability of metastasis and should by deeply discussed.

2)     Figure 2  - analysis of EVs markers should consist not only of EVs and BEAS-2B cells, but also MCF10a, MDA-MB-231, and HS578T cell lysates as a control to the isolated EVs. Why did authors not try to use any other tetraspanin (for example CD9) or any other EVs marker in a situation where EVs isolated from MCF10a cells present no western blot detection?

3)     LINE 175-177: “Specifically, EVs derived from MCF10a-cells showed a lower expression of two miRNAs (miR-185-5p and miR-652-5p), known to be involved in cell migration, compared to EVs from TNBC cells, which suggest a more aggressive effect.” Is this the author's data? No references are given.

4)     According to the doi.org/10.1038/s41598-023-35310-5 and doi.org/10.1016/j.tiv.2013.04.012 doubling time for BEAS-2B cell lines is 22.5-26h. I do not understand why the authors performed a wound healing assay in the presence of FBS and for the period of 72h? In these conditions, the effect of cell motility is eclipsed by their proliferation. This fact seems to be known by the authors as they state: “Additionally, between the control and cells with MCF10a EVs we didn’t observe much differences, suggesting that breast cancer derived EVs stimulate cell proliferation and growth”. Additionally, (in my opinion) 24 or 12 well plates would be better choice than 6 well plates for this experiment, as the area of the artificially created wound, would comprise of a higher percentage of the overall cell surface, thus the cell proliferation effect would be less pronounced (not mentioning 0% FBS media or proliferation inhibitors). If proliferation was the main phenomenon to be analyzed in this section than different methods should be used.

5)     The article seems to present preliminary data and does not try to explain observed mechanisms.

Minor issues:

1)     In my opinion this article requires proper English correction.

2)     I doubt that we can use the term “aggressiveness” regarding normal epithelial lung cells.

3)     LINE 25-26: “Breast cancer is one of the most common cancers in the world and mainly affects women, but new cases are also occurring among men” – this sentence should be rephrased. Less than 1% of lung cancer instances are observed among men, but stating that “new cases are also occurring among men” suggests that earlier no cases were observed or were not occurring among men at all.

4)     Please correct Figure 1. “Particle concentration” and “Particle size” do not align.

5)     Please rephrase LINE 52-53: “Thus, EVs may be possible biomarkers to be targeted for preventing metastatic development” – “Biomarkers” are rather indicators not therapeutical targets per se.

6)     LINE 85-86: “Specifically, three EVs associated protein were used, TSG101, CD81, CD63 and one no EVs-associated marker” – from figure 2 I understand that it refers to Calnexin. It should be mentioned in the text.

The strong side of the manuscript:

It is worth noting, that authors provided EVs depleted FBS for their experiments and used NTA system to verify isolated EVs. Additionally, they have analyzed EVs uptake by recipient cells.

Comments on the Quality of English Language

This manuscript requires some moderate English editing. Several sentences use inadequate terms, others have some grammatical and/or structural issues. Additionally sentences such as: “EVs associated protein were used” should be changed into: EVs-associated proteins were used… and many others small mistakes.

Author Response

REVIEWER 1

Open Review

Quality of English Language

( ) I am not qualified to assess the quality of English in this paper
( ) English very difficult to understand/incomprehensible
( ) Extensive editing of English language required
(x) Moderate editing of English language required
( ) Minor editing of English language required
( ) English language fine. No issues detected

Yes

Can be improved

Must be improved

Not applicable

Does the introduction provide sufficient background and include all relevant references?

( )

(x)

( )

( )

Are all the cited references relevant to the research?

(x)

( )

( )

( )

Is the research design appropriate?

( )

( )

(x)

( )

Are the methods adequately described?

( )

( )

(x)

( )

Are the results clearly presented?

( )

(x)

( )

( )

Are the conclusions supported by the results?

( )

( )

(x)

( )

Comments and Suggestions for Authors

Breast cancer is one of the most common cancer types observed in women's worldwide population (with only approximately less than 1% of men affected). Despite increased diagnostic capabilities and therapeutic schemes, breast cancer and its metastasis to the lungs, bones, brain, and liver remains a huge problem for current oncology. The article titled “Triple Negative Breast Cancer EVs Modulate Growth and 2 Migration of Normal Epithelial Lung Cells” by Leone I. et al. focuses on TNBC-derived EVs impact on normal lung cell physiology and its potential connotation with preparation of the pre-metastatic niche for breast cancer metastasis. The authors addressed this issue by isolating EVs from two TNBC (MDA-MB-231 and HS578T) cell lines and analyzing their impact on BEAS-2B cell line migration. This is an interesting subject, however, in my opinion, the current version requires substantial revision, as it seems to present rather preliminary data and lacks some important parts (and possibly some concluding experiments) to be published by IJMS.

My concerns can be distinguished into major and minor.

Major issues:

  • In my opinion this manuscript lacks proper discussion of the obtained results. The “Results and Discussion” section mainly focuses on the acquired results but not on their meaning. No mechanism for increased migration of normal lung cells is proposed, absence of CD81 and CD63 in the EVs isolated from MCF10a cells is not discussed at all. Statements such as: (LINE 289-292) “We believe that our data will help in understanding the transition to the metastatic stage. We provide evidence for the use of extracellular vesicles as early biomarker detectable in several fluids with the objective of predicting metastasis” which are not backed by the acquired data nor discussed. Increased migration of normal lung cells does not directly contribute to the increased probability of metastasis and should by deeply discussed.

We thank the Reviewer for this thoughtful comment. We have now rearranged and improved the “Results and discussion” section.

  • Figure 2 - analysis of EVs markers should consist not only of EVs and BEAS-2B cells, but also MCF10a, MDA-MB-231, and HS578T cell lysates as a control to the isolated EVs. Why did authors not try to use any other tetraspanin (for example CD9) or any other EVs marker in a situation where EVs isolated from MCF10a cells present no western blot detection?

We thank the Reviewer for this comment. We amended the immunoblotting results by discussing extensively the markers observed in our study. We used BEAS-2B cells as positive control for the detection of EVs markers, which supported our results on the effect of TNBC EVs on normal epithelial lung cells. We are aware that in MCF10a EVs some markers were not readily detected (except for TSG101 and CD63), however for all the rest of the EVs samples, EVs markers were clearly detected. Considering that, for future studies we will implement our EVs characterization with other EVs markers in line with the MISEV2023 guidelines.

LINE 175-177: “Specifically, EVs derived from MCF10a-cells showed a lower expression of two miRNAs (miR-185-5p and miR-652-5p), known to be involved in cell migration, compared to EVs from TNBC cells, which suggest a more aggressive effect.” Is this the author's data? No references are given.

We thank the Reviewer for this comment. We have now added the appropriate reference for the cited work (Line 209-211) in the manuscript.

4)     According to the doi.org/10.1038/s41598-023-35310-5 and doi.org/10.1016/j.tiv.2013.04.012 doubling time for BEAS-2B cell lines is 22.5-26h. I do not understand why the authors performed a wound healing assay in the presence of FBS and for the period of 72h? In these conditions, the effect of cell motility is eclipsed by their proliferation. This fact seems to be known by the authors as they state: “Additionally, between the control and cells with MCF10a EVs we didn’t observe much differences, suggesting that breast cancer derived EVs stimulate cell proliferation and growth”. Additionally, (in my opinion) 24 or 12 well plates would be better choice than 6 well plates for this experiment, as the area of the artificially created wound, would comprise of a higher percentage of the overall cell surface, thus the cell proliferation effect would be less pronounced (not mentioning 0% FBS media or proliferation inhibitors). If proliferation was the main phenomenon to be analyzed in this section than different methods should be used.

      We thank the Reviewer for this thoughtful comment. We performed the wound healing assay in presence of FBS 10% according to Scognamiglio et al. (2022). Moreover, we tested the wound closure up to 72h to observe this phenomenon also in presence of EVs from normal cell line (MCF10a). As suggested, for Line 157, we changed the word “proliferation” with “motility” to better describe this effect.

5)     The article seems to present preliminary data and does not try to explain observed mechanisms.

We thank the Reviewer for the comment and we agree with the Reviewer. Therefore, we rearranged this version of the manuscript by deeply discussing our results. Although, we are aware that our work is only preliminary, we hope that our study might shed a light on new approach for early cancer detection, such as the use of EVs as diagnostic tool. In fact, future studies will investigate the role of TNBC EVs in tumor progression and their activity using high-throughput analysis.

Minor issues:

1)     In my opinion this article requires proper English correction.

       We thank the Reviewer for the comment. We improved the English of the manuscript.

2)     I doubt that we can use the term “aggressiveness” regarding normal epithelial lung cells.

      We thank the Reviewer for the comment. We have now changed the term “aggressiveness” with an appropriate sentence in the conclusion section of the manuscript. 

3)     LINE 25-26: “Breast cancer is one of the most common cancers in the world and mainly affects women, but new cases are also occurring among men” – this sentence should be rephrased. Less than 1% of lung cancer instances are observed among men, but stating that “new cases are also occurring among men” suggests that earlier no cases were observed or were not occurring among men at all.

      We thank the Reviewer for this suggestion. We have now corrected the Line 25-26 with: “Breast cancer is one of the most common cancers in the world and mainly affects women, but occasionally they are also occurring among men”

 4)     Please correct Figure 1. “Particle concentration” and “Particle size” do not align.

       We thank the Reviewer for this comment. We have now modified Fig.1 by aligning “Particle concentration” and “Particle size”, in the manuscript, as suggested.

5)     Please rephrase LINE 52-53: “Thus, EVs may be possible biomarkers to be targeted for preventing metastatic development” – “Biomarkers” are rather indicators not therapeutical targets per se.

We thank the Reviewer for the suggestion. We have now rephrased the Line 55-56 “Thus, EVs may be possible biomarkers to be targeted for preventing metastatic development” with “Thus, EVs may be used as reliable biomarkers for early cancer detection and cancer development”.

6)     LINE 85-86: “Specifically, three EVs associated protein were used, TSG101, CD81, CD63 and one no EVs-associated marker” – from figure 2 I understand that it refers to Calnexin. It should be mentioned in the text.

We thank the Reviewer for this comment. We specified the marker “Calnexin” in the manuscript. It is reported now “Specifically, three EVs associated protein were used, TSG101, CD81, CD63 and one no EVs-associated marker calnexin”, as showed in Fig. 2.”

The strong side of the manuscript:

It is worth noting, that authors provided EVs depleted FBS for their experiments and used NTA system to verify isolated EVs. Additionally, they have analyzed EVs uptake by recipient cells.

We thank the Reviewer for the thoughtful comment.

Comments on the Quality of English Language

This manuscript requires some moderate English editing. Several sentences use inadequate terms, others have some grammatical and/or structural issues. Additionally sentences such as: “EVs associated protein were used” should be changed into: EVs-associated proteins were used… and many others small mistakes.

We thank the Reviewer for these suggestions. We corrected “EVs associated protein” with “EVs-associated proteins”, as suggested. Furthermore, a comprehensive linguistic edit of the manuscript was made, grammatical and syntax errors scattered throughout the manuscript were corrected.

Reviewer 2 Report

Comments and Suggestions for Authors

The study investigates how extracellular vesicles (EVs) from triple negative breast cancer cells (TNBC) act on lung epithelial cells (BEAS-2B). Using EVs isolated from the TNBC cell lines MDA-MB-231 and HS578T, the researchers treated BEAS-2B cells and monitored their growth and migration. The results indicate that TNBC-derived EVs enhance the pro-tumorigenic behavior of these lung cells, suggesting that cargo in tumor-derived EVs has the potential for novel biomarkers and pathogenetic mechanisms of the oncologic process. There are a number of comments on the manuscript, which are listed below.

In order to accurately compare the number of extracellular vesicles between different cell cultures, normalization of the final cell numbers is essential. This adjustment takes into account possible discrepancies in cell growth rates that could affect vesicle production.

For Figure 1 “Particle size”, it is necessary to represent the entire spectrum of particle size distribution instead of considering only selected particles and performing a statistical population analysis.

The results presented in sections “2.2 EV Marker Analysis by Immunoblotting” and “2.1 Analysis of particle concentration and size using nanoparticle tracking analysis” could be merged. This would streamline the description of EV characterization results and reflect their routine application in this area of research.

The manuscript would benefit from an examination of the dose-dependent toxic effects of vesicles. The rationale for the choice of specific concentration for toxicity studies and the lack of a dose-response analysis require further clarification.

To convincingly demonstrate the intracellular localization of EVs in the “Evaluation of EV Uptake through Confocal Microscopy” experiment, the z-axis needs to be included. This allows a more detailed visualization of EV distribution within the cell cytoplasm.

The use of a consistent dose of 10^8 EV vesicles in both the scratch assays and the transwell migration assay studies raises questions about the consistency of vesicle concentration, despite the different volumes of the culture plates. An explanation of how these conditions were standardized or accounted for would be beneficial.

A critical drawback of this article is the lack of discussion of the data obtained and its comparison with work in this area. It also creates confusion about what kind of “aggressiveness of BEAS-2B cells” we can talk about if this cell line is a non-tumorigenic epithelial cell line from human bronchial epithelium.

Author Response

REVIEWER 2

Open Review

Quality of English Language

(x) I am not qualified to assess the quality of English in this paper
( ) English very difficult to understand/incomprehensible
( ) Extensive editing of English language required
( ) Moderate editing of English language required
( ) Minor editing of English language required
( ) English language fine. No issues detected

Yes

Can be improved

Must be improved

Not applicable

Does the introduction provide sufficient background and include all relevant references?

( )

(x)

( )

( )

Are all the cited references relevant to the research?

(x)

( )

( )

( )

Is the research design appropriate?

( )

( )

(x)

( )

Are the methods adequately described?

( )

(x)

( )

( )

Are the results clearly presented?

( )

( )

(x)

( )

Are the conclusions supported by the results?

( )

( )

(x)

( )

Comments and Suggestions for Authors

The study investigates how extracellular vesicles (EVs) from triple negative breast cancer cells (TNBC) act on lung epithelial cells (BEAS-2B). Using EVs isolated from the TNBC cell lines MDA-MB-231 and HS578T, the researchers treated BEAS-2B cells and monitored their growth and migration. The results indicate that TNBC-derived EVs enhance the pro-tumorigenic behavior of these lung cells, suggesting that cargo in tumor-derived EVs has the potential for novel biomarkers and pathogenetic mechanisms of the oncologic process. There are a number of comments on the manuscript, which are listed below.

We thank the Reviewer for the thoughtful comments.

In order to accurately compare the number of extracellular vesicles between different cell cultures, normalization of the final cell numbers is essential. This adjustment takes into account possible discrepancies in cell growth rates that could affect vesicle production.

We thank the Reviewer for this comment. We have now updated the particles concentration graph in Fig.1a according to the final cell numbers, as suggested.

For Figure 1 “Particle size”, it is necessary to represent the entire spectrum of particle size distribution instead of considering only selected particles and performing a statistical population analysis.

We thank the Reviewer for this comment. For the current study we considered the overall particles size of our EVs preparation. As we did not evaluate the different EVs subpopulations, here, the graphs reported in Fig.1 are representative of EVs which are in the 30-150 nm size range.

The results presented in sections “2.2 EV Marker Analysis by Immunoblotting” and “2.1 Analysis of particle concentration and size using nanoparticle tracking analysis” could be merged. This would streamline the description of EV characterization results and reflect their routine application in this area of research.

We thank the Reviewer for this comment. As suggested, we merged the paragraphs and related figures of “2.1 Analysis of particle concentration and size using nanoparticle tracking analysis” and “2.2 EV Marker Analysis by Immunoblotting” into one. It is now rearranged as “2.1. Characterization of EVs from TNBC and MCF10a cell lines by NTA and Immunoblotting Analysis.”

The manuscript would benefit from an examination of the dose-dependent toxic effects of vesicles. The rationale for the choice of specific concentration for toxicity studies and the lack of a dose-response analysis require further clarification.

We thank the Reviewer for the comment. Prior to perform our experiment in this study, we previously tested different EVs concentration (data not shown) in order to choose the suitable concentration.

To convincingly demonstrate the intracellular localization of EVs in the “Evaluation of EV Uptake through Confocal Microscopy” experiment, the z-axis needs to be included. This allows a more detailed visualization of EV distribution within the cell cytoplasm.

We thank the Reviewer for this suggestion. We did not include the z-axis because we used the Maximum Projection to obtain a better image. Accordingly, we corrected the caption of Fig. 4 in the manuscript as “Maximum projection images of BEAS-2B cells incubated with 108 EVs separated from MDA-MB-231, HS578T and MCF10a cell lines”.

The use of a consistent dose of 10^8 EV vesicles in both the scratch assays and the transwell migration assay studies raises questions about the consistency of vesicle concentration, despite the different volumes of the culture plates. An explanation of how these conditions were standardized or accounted for would be beneficial.

We thank the Reviewer for this thoughtful comment. Regarding the unit used in sections “3.7. Scratch Assay” and "3.8. Transwell Migrations assays", we referred to integer of EVs (108) and not to particles concentration/mL.

A critical drawback of this article is the lack of discussion of the data obtained and its comparison with work in this area. It also creates confusion about what kind of “aggressiveness of BEAS-2B cells” we can talk about if this cell line is a non-tumorigenic epithelial cell line from human bronchial epithelium.

      We thank the Reviewer for this comment. We have now improved the Results and Discussion section of the manuscript. Furthermore, we rearranged the conclusion part in order to better summaries our data and to avoid any confusion for the readers. Moreover, we apologize if we were not quite clear in our discussion of the results obtained in this study and we hope that it is now fluently described.

Round 2

Reviewer 1 Report

Comments and Suggestions for Authors

The majority of my comments were addressed, however, the two most important ones were not properly answered.

Figure 2 authors did not provide western blots of MCF10a, MDA-MB-231, and HS578T cell lysates as a control to the isolated EVs. In my opinion, such control should be provided as EVs were isolated from these cells. However, this issue may be neglected as it is not as severe as the problem with wound healing/motility analysis. The authors refer to the wound healing assay reported by Scognamiglio et al. (2022) (https://doi.org/10.1016/j.omtn.2022.02.013) regarding the usage of the FBS-supplemented cell medium: “(…) NFs (5 × 104) were seeded in a 12-well plate (Corning, 3513) and, on the following day, transfected with miR-185-5p, miR-652-5p, miR-1246, or a scrambled sequence as a control. After 48 h, cells were starved for 3 h in DMEM-F12 FBS-free culture medium. Next, a scratched wound was made with a 200-μL tip in each well, and then cells were grown continuously in DMEM-F12 culture medium complemented with 10% FBS and 1% A/A for 24 h. Microscopy images were taken in different fields of the wound at the scratch moment (t0) and after 24 h (t24) using a 5× objective of an inverted microscope (DMI3000 B, Leica, Milan, Italy). (…)“. Nevertheless, the main differences were misinterpreted, Scognamiglio et al. performed wound healing using a wound enclosure time of 24h, and using primary fibroblast cells isolated from patients. Primary cells' ability to proliferate is significantly lower than commercially available immortalized cell lines, thus 10% FBS supplementation would not affect the observed motility in the mentioned 24-hour period. In the case of the reviewed manuscript in which the BEAS-2B cell line is used, proliferation in 10% FBS-supplemented medium would impact the wound enclosure in a much greater scale than cells increased motility (especially after 24 hours). This experiment should be repeated in the correct way (excluding proliferation, either by starving conditions via FBS depletion, or using proliferation inhibitors) and if the authors wish to emphasize the impact of TNBC-derived EVs on cell proliferation, then correct test (such as doubling time) should be performed.

Author Response

Quality of English Language

( ) I am not qualified to assess the quality of English in this paper
( ) English very difficult to understand/incomprehensible
( ) Extensive editing of English language required
( ) Moderate editing of English language required
( ) Minor editing of English language required
(x) English language fine. No issues detected

Yes

Can be improved

Must be improved

Not applicable

Does the introduction provide sufficient background and include all relevant references?

( )

(x)

( )

( )

Are all the cited references relevant to the research?

(x)

( )

( )

( )

Is the research design appropriate?

( )

( )

(x)

( )

Are the methods adequately described?

( )

( )

(x)

( )

Are the results clearly presented?

( )

(x)

( )

( )

Are the conclusions supported by the results?

( )

( )

(x)

( )

Comments and Suggestions for Authors

The majority of my comments were addressed; however, the two most important ones were not properly answered.

Figure 2 authors did not provide western blots of MCF10a, MDA-MB-231, and HS578T cell lysates as a control to the isolated EVs. In my opinion, such control should be provided as EVs were isolated from these cells. However, this issue may be neglected as it is not as severe as the problem with wound healing/motility analysis.

We thank the Reviewer for this comment. As suggested, we have now added the western blot of the cells used for EVs separation and it was added to Fig. S1. The updated Figure S1 with uncropped WB images was deposited in the Supplementary material file.

The authors refer to the wound healing assay reported by Scognamiglio et al. (2022) (https://doi.org/10.1016/j.omtn.2022.02.013) regarding the usage of the FBS-supplemented cell medium: “(…) NFs (5 × 104) were seeded in a 12-well plate (Corning, 3513) and, on the following day, transfected with miR-185-5p, miR-652-5p, miR-1246, or a scrambled sequence as a control. After 48 h, cells were starved for 3 h in DMEM-F12 FBS-free culture medium. Next, a scratched wound was made with a 200-μL tip in each well, and then cells were grown continuously in DMEM-F12 culture medium complemented with 10% FBS and 1% A/A for 24 h. Microscopy images were taken in different fields of the wound at the scratch moment (t0) and after 24 h (t24) using a 5× objective of an inverted microscope (DMI3000 B, Leica, Milan, Italy). (…)“. Nevertheless, the main differences were misinterpreted, Scognamiglio et al. performed wound healing using a wound enclosure time of 24h, and using primary fibroblast cells isolated from patients. Primary cells' ability to proliferate is significantly lower than commercially available immortalized cell lines, thus 10% FBS supplementation would not affect the observed motility in the mentioned 24-hour period. In the case of the reviewed manuscript in which the BEAS-2B cell line is used, proliferation in 10% FBS-supplemented medium would impact the wound enclosure in a much greater scale than cells increased motility (especially after 24 hours). This experiment should be repeated in the correct way (excluding proliferation, either by starving conditions via FBS depletion, or using proliferation inhibitors) and if the authors wish to emphasize the impact of TNBC-derived EVs on cell proliferation, then correct test (such as doubling time) should be performed.

We thank the Reviewer for this comment. We are aware that other groups performed similar experiment by using inhibitors or starving conditions in order to confirm that the motility of BEAS-2B cells was potentially driven by cancer EVs. However, in this study we used medium supplemented with FBS-EVs depleted, which was made by ultracentrifugation (18h at 110,000 x g). In fact, it was demonstrated (https://isevjournals.onlinelibrary.wiley.com/doi/10.3402/jev.v3.24783) that FBS EVs have a direct migratory effect on lung cancer cells (A549), this effect is significantly reduced when FBS is subjected to extended centrifugation, which is commonly utilized to remove FBS EVs. Therefore, based on that, we performed our experiment using FBS after 18h at 110,000 g is potentially deprived of EVs and some nutrients as well. Moreover, this study demonstrated that overnight centrifugation removes most of the FBS-mediated effect on cell migration.

Reviewer 2 Report

Comments and Suggestions for Authors

The authors addressed only a portion of my feedback. Unfortunately, the revised article still retains the significant flaws that were highlighted in my original review.

Author Response

REVIEWER 2

Open Review

Quality of English Language

(x) I am not qualified to assess the quality of English in this paper
( ) English very difficult to understand/incomprehensible
( ) Extensive editing of English language required
( ) Moderate editing of English language required
( ) Minor editing of English language required
( ) English language fine. No issues detected

Yes

Can be improved

Must be improved

Not applicable

Does the introduction provide sufficient background and include all relevant references?

( )

(x)

( )

( )

Are all the cited references relevant to the research?

(x)

( )

( )

( )

Is the research design appropriate?

( )

( )

(x)

( )

Are the methods adequately described?

( )

(x)

( )

( )

Are the results clearly presented?

( )

( )

(x)

( )

Are the conclusions supported by the results?

( )

( )

(x)

( )

Comments and Suggestions for Authors

The study investigates how extracellular vesicles (EVs) from triple negative breast cancer cells (TNBC) act on lung epithelial cells (BEAS-2B). Using EVs isolated from the TNBC cell lines MDA-MB-231 and HS578T, the researchers treated BEAS-2B cells and monitored their growth and migration. The results indicate that TNBC-derived EVs enhance the pro-tumorigenic behavior of these lung cells, suggesting that cargo in tumor-derived EVs has the potential for novel biomarkers and pathogenetic mechanisms of the oncologic process. There are a number of comments on the manuscript, which are listed below.

We thank the Reviewer for the thoughtful comments.

In order to accurately compare the number of extracellular vesicles between different cell cultures, normalization of the final cell numbers is essential. This adjustment takes into account possible discrepancies in cell growth rates that could affect vesicle production.

We thank the Reviewer for this comment. We have adjusted the particles concentration according to the final number of the cells (see figure below). The updated graph of EVs concentration and statistical differences are reported in Fig.1 in the manuscript.

For Figure 1 “Particle size”, it is necessary to represent the entire spectrum of particle size distribution instead of considering only selected particles and performing a statistical population analysis.

We thank the Reviewer for this comment. For the current study we considered the overall particles size of our EVs preparation. As we did not evaluate the different EVs subpopulations, here, the graphs reported in Fig.1a are representative of the obtained EVs after differential ultracentrifugation which were in the 30-150 nm size range as shown in Fig 1a. Additionally, we added the representative size distribution graphs of EVs (see figure below) observed with the NTA software in the Supplementary material, as Fig.S3.

The results presented in sections “2.2 EV Marker Analysis by Immunoblotting” and “2.1 Analysis of particle concentration and size using nanoparticle tracking analysis” could be merged. This would streamline the description of EV characterization results and reflect their routine application in this area of research.

We thank the Reviewer for this comment. As suggested, we merged the paragraphs and related figures of “2.1 Analysis of particle concentration and size using nanoparticle tracking analysis” and “2.2 EV Marker Analysis by Immunoblotting” into one. It is now rearranged as “2.1. Characterization of EVs from TNBC and MCF10a cell lines by NTA and Immunoblotting Analysis.”

The manuscript would benefit from an examination of the dose-dependent toxic effects of vesicles. The rationale for the choice of specific concentration for toxicity studies and the lack of a dose-response analysis require further clarification.

We thank the Reviewer for the comment. Prior to performing our experiment in this study, we previously performed scratch assay testing different EVs concentration, as reported in the figure below, separated from MDA-MB-231 cells, in order to choose the most effective EVs concentration for our purpose. By testing different concentration of EVs, we observed that EVs at 108 caused the maximum wound closure. For this reason, we decided to use EVs at for all 108 the experiments.

In this regard, it was reported in literature (https://link.springer.com/article/10.1007/s10549-018-4925-5) that no effects were observed by the treatment with EVs from the MDA-MB-231 cell line on recipient cells after 48h.

To convincingly demonstrate the intracellular localization of EVs in the “Evaluation of EV Uptake through Confocal Microscopy” experiment, the z-axis needs to be included. This allows a more detailed visualization of EV distribution within the cell cytoplasm.

We thank the Reviewer for this suggestion. We did not include the z-axis because we used the Maximum Projection to obtain a better image. Accordingly, we corrected the caption of Fig. 4 in the manuscript as “Maximum projection images of BEAS-2B cells incubated with 108 EVs separated from MDA-MB-231, HS578T and MCF10a cell lines”.

The use of a consistent dose of 10^8 EV vesicles in both the scratch assays and the transwell migration assay studies raises questions about the consistency of vesicle concentration, despite the different volumes of the culture plates. An explanation of how these conditions were standardized or accounted for would be beneficial.

We thank the Reviewer for this thoughtful comment. Regarding the unit used for the scratch and transwell migration assays are referring to integer of EVs (108) and not to particles concentration/mL. In fact, another study also used the same integer of EVs to perform functional studies (https://www.mdpi.com/2079-7737/12/12/1531). Moreover, the results obtained with these assays using this approach are further discussed in the revised manuscript.

A critical drawback of this article is the lack of discussion of the data obtained and its comparison with work in this area. It also creates confusion about what kind of “aggressiveness of BEAS-2B cells” we can talk about if this cell line is a non-tumorigenic epithelial cell line from human bronchial epithelium.

      We thank the Reviewer for this comment. We improved the manuscript, based on the comments suggested by the Reviewer. Particularly, we focused on the discussion of the obtained results which showed that TNBC EVs significantly incremented growth and proliferation of normal human bronchial epithelial cells when treated with breast cancer EVs which was not observed in the case of EVs derived from MFC10a cell line, where no significant differences were observed compared to the control. Therefore, our results propose potential molecular signatures of the studied TNBC cell lines and their impact in growth and proliferation rates. Furthermore, we rearranged the conclusion part in order to meet the suggestions reported by the Reviewer and to avoid any confusion for the readers. Moreover, we apologize if we were not quite clear in our discussion of the results obtained in this study and we hope that it is now fluently described.

Round 3

Reviewer 1 Report

Comments and Suggestions for Authors

I appreciate the western blot of missing cell lysates. However, I feel that the authors did not understand my concerns regarding FBS supplementation during wound healing experiments. I noticed the fact that FBS was depleted of any EVs via ultracentrifugation, but the main problem involves cell proliferation. Cells that were seeded on a multi-well plate for the first few hours are in “the lag phase” and adapt to the new environment stopping their proliferation. Next, cells resume proliferation if supplemented by FBS, thus 4according to the doi.org/10.1038/s41598-023-35310-5 and doi.org/10.1016/j.tiv.2013.04.012, BEAS-2B cell doubles their number in every 22.5-26h. To make it easier let's assume that the doubling time of BEAS-2B  is 24h, and the lag phase took also 24h, and wound healing is performed using 2 million cells per well. If one seeds 2 million cells after 24h of lag phase will have 2 million cells, and the next 24h after performing the scratch in FBS supplemented medium one will have 2x2 million cells, which equals 4 million. Next 24h (48h since the scratch) and 2x4 million, which equals 8 million, and finally 72h from scratch one will receive 16 million cells. This is just an approximation, as cells will not continue dividing using a logarithmic scale for a prolonged time, but it gives the idea of cell proliferation impact on wound healing assay and the necessity to use either proliferation inhibitors or starving conditions. Otherwise, dividing cells will be urged to find a spot and, thus will be pushing one another and this effect will directly impact wound enclosure and may substantially shadow the analyzed effect of EVs impact. This is even more visible in cases where the wound area constitutes of small percent of the overall cell area, which means that it is better to use 24 or 12 well plates and a bigger pipet tip, than 6 well plates and tiny scratches.

I believe the authors that, TNBC-derived EVs enhance BEAS-2B motility and invasiveness (proven using trans-well assay), however, it can't be presented using an incorrectly designed wound healing assay, which might influence other authors to repeat and consolidate this mistake. This experiment must be repeated. Additionally, if the authors insist on proving that TNBC-derived EVs enhance BEAS-2B cell proliferation, they should use a different method.

Reviewer 2 Report

Comments and Suggestions for Authors

The manuscript presented in this study raises serious concerns about the inconsistencies in the results obtained with different approaches. In particular, the authors found discrepancies between the results of the MTT analysis, which showed no effect of extracellular vesicles on cell proliferation, and the results of the wound healing assay, which showed an increase in proliferation. These discrepancies need to be addressed and discussed to clarify the validity of the results.

In addition, the authors are advised to separate the "Discussion" section in order to thoroughly discuss and dissect the results obtained. Is it believed that cell physiology in the lungs of women with triple-negative cancer may be altered, or is this an artifact of in vitro research?

A discussion is required about what mechanisms extracellular vesicles can use to achieve the observed effects in the article on the effect on the cells of the pulmonary epithelium.

In conclusion, careful consideration and discussion of the contradictory results, as well as an exploration/discussion of the mechanisms underlying the observed effects of extracellular vesicles, will enhance the scientific rigor and impact of this study.